# WIN55,212-2, a Dual Modulator of Cannabinoid Receptors and G Protein-Coupled Inward Rectifier Potassium Channels

**DOI:** 10.3390/biomedicines9050484

**Published:** 2021-04-28

**Authors:** Dongchen An, Steve Peigneur, Jan Tytgat

**Affiliations:** Toxicology and Pharmacology, KU Leuven, Campus Gasthuisberg, O & N2, Herestraat 49, P.O. Box 922, 3000 Leuven, Belgium; dongchen.an@kuleuven.be

**Keywords:** dual modulation, WIN55,212-2, cannabinoid receptor type 1 (CB1) and type 2 (CB2), G protein-coupled inward rectifier potassium channels 1 (GIRK1) and 2 (GIRK2)

## Abstract

The coupling of cannabinoid receptors, CB1 and CB2, to G protein-coupled inward rectifier potassium channels, GIRK1 and GIRK2, modulates neuronal excitability in the human brain. The present study established and validated the functional expression in a *Xenopus laevis* oocyte expression system of CB1 and CB2 receptors, interacting with heteromeric GIRK1/2 channels and a regulator of G protein signaling, RGS4. This *ex vivo* system enables the discovery of a wide range of ligands interacting orthosterically or allosterically with CB1 and/or CB2 receptors. WIN55,212-2, a non-selective agonist of CB1 and CB2, was used to explore the CB1- or CB2-GIRK1/2-RGS4 signaling cascade. We show that WIN55,212-2 activates CB1 and CB2 at low concentrations whereas at higher concentrations it exerts a direct block of GIRK1/2. This illustrates a dual modulatory function, a feature not described before, which helps to explain the adverse effects induced by WIN55,212-2 *in vivo*. When comparing the effects with other typical cannabinoids such as Δ9-THC, CBD, CP55,940, and rimonabant, only WIN55,212-2 can significantly block GIRK1/2. Interestingly, the inward rectifier potassium channel, IRK1, a non-G protein-coupled potassium channel important for setting the resting membrane voltage and highly similar to GIRK1 and GIRK2, is not sensitive to WIN55,212-2, Δ9-THC, CBD, CP55,940, or rimonabant. From this, it is concluded that WIN55,212-2 selectively blocks GIRK1/2.

## 1. Introduction

Cannabinoid receptors (CB) are G protein-coupled receptors (GPCRs) that have attracted broad attention since they are regarded as promising therapeutic targets for multiple pathologies, such as pain, epilepsy, anorexia, Parkinson’s disease, and Alzheimer’s disease [1]. Cannabinoid receptors are primarily classified as CB1 and CB2. CB1 is one of the most abundant GPCRs in the human brain [2], while CB2 was originally considered to be confined in the peripheral organs of humans [3,4]. Afterward, the expression of CB2 has also been reported in the mammalian healthy central nervous system (CNS) [5,6,7,8] and diseased brain cells from humans [9,10,11]. Ligands of CB1 and/or CB2 (also commonly known as cannabinoids), through their binding to cannabinoid receptors in the brain, produce a wide variety of effects. For instance, activating CB1 generates psychotropic effects including euphoria, enhancement of sensory perception, antinociception, appetite stimulation, and impairment of memory [1]. Activating CB2 produces antinociception, antiinflammation, and neuroprotection [1]. In the brain, signaling pathways of activating CB1 and/or CB2 have been revealed involving the phosphorylation of receptors by G protein receptor kinases and subsequently an association with β-arrestin1 or β-arrestin2 resulting in the desensitization and internalization of cannabinoid receptors [2,12]. Besides, activation of CB1 and/or CB2 elicits the dissociation of the βγ subunits (Gi/o(βγ)) of pertussis toxin-sensitive G proteins (Gi/o) from the α subunits (Gi/o(α)) [2,12,13]. Gi/o(α) inhibits adenylyl cyclase, thereby reducing cellular cAMP levels [12]. Both Gi/o(α) and β-arrestin can also stimulate different members of the mitogen-activated protein kinases family, bringing about additional cellular effects [2,12,13]. On the other hand, Gi/o(βγ) modulates the activity of several ion channels. Gi/o(βγ) inhibits the opening of Ca^2+^ channels (N & P/Q type) and opens G protein-coupled inward rectifier K^+^ (GIRK) channels [12]. The GIRK channels are composed of four mammalian subunits, GIRK1 (Kir3.1), GIRK2 (Kir3.2), GIRK3 (Kir3.3), and GIRK 4 (Kir3.4), and are members of a family of inward rectifier K^+^ (Kir) channels [14,15]. GIRK1-3 are predominant subunits in the brain with overlapping but distinct expression patterns throughout the CNS and peripheral nervous system that form heterotetrameric channels [16]. Notably, GIRK1/2 is the most common heterotetrameric GIRK subunit arrangement expressed in the brain [17,18,19].

Heterotetramers of GIRK subunits exist as primary functional GIRK channels that regulate neuronal excitability in the brain [14,16]. The activation of GIRK channels hyperpolarizes the membrane potential of neurons, and thereby reduces action potential firing [20]. Thus, they inhibit the release of many neurotransmitters including dopamine [16,17,19]. GIRK activation also improves hippocampal inhibitory activity disrupted by amyloid-β (Aβ), which indicates that GIRK channels play an important role in the restoration of the excitatory and inhibitory balance impaired by Aβ [21]. This could prevent neuronal dysfunction and the cognitive deficits associated with the early stages of Alzheimer’s disease [21]. Moreover, aberrant GIRK activity throughout the brain has been implicated in pain, epilepsy, drug addiction, deficits in learning and memory, Down’s Syndrome, and Parkinson’s disease [17]. Therefore, GIRK channels indeed play important roles in both health and disease.

The binding of several cannabinoids to CB1 and/or CB2 activates the G-proteins-mediated signaling cascade and then produces coordinated changes in several cellular effector systems including triggering GIRK channels [13,22]. For example, Δ9-tetrahydrocannabinol (Δ9-THC) and CP55,940 are more potent mediators of β-arrestin2 recruitment than other agonists, whereas anandamide and WIN55, 212-2 prefer to stimulate Gi/o pathways [22,23,24]. WIN55,212-2, one of the representatives of the non-selective agonists of CB1 and CB2, is a synthetic cannabinoid that can exert analgesic, anti-inflammatory, and neuroprotective bioactivities *in vivo* [1]. Despite the promising prospects, further clinical uses of WIN55,212-2 are precluded due to its severe side effects, such as anxiety [25], recognition memory impairment, and brain network functional connectivity impairment [26]. Nevertheless, WIN55,212-2 has been widely utilized as an experimental tool to explore the mechanism of the cannabinoid system or cannabinoid-mediated signaling and to study structure-receptor activity relationships to inspire new classes of cannabinoids. It has been reported to enhance Gi/o (α) and Gi/o(βγ) signaling after binding to CB1 in the mouse STHdh(Q7/Q7) cell culture model of striatal medium spiny projection neurons that endogenously express CB1 [24]. Additionally, the application of WIN55,212-2 to AtT20 pituitary cells expressing CB1 activated GIRK currents [15]. In a *Xenopus* oocyte system co-expressing CB1 and GIRK1/4, WIN55,212-2 enhanced currents carried by GIRK1/4 through activating CB1 [27]. CB2, however, did not couple efficiently to GIRK1/4 in their system [27]. Further evidence shows that the analgesic effects of WIN55,212-2 were reduced or eliminated in male GIRK2^−/−^ mice [17]. Based on our knowledge, the characterization of the effects of WIN55,212-2 on CB2 is still unknown.

Thus, this study aims to further explore the interactions between WIN55,212-2 and cannabinoid receptors coupling to GIRK channels, particularly the GIRK1/2 expressed in the human brain. First of all, we built and validated a reliable *Xenopus* oocyte system efficiently co-expressing CB1 or CB2, GIRK1/2, and a regulator of the G protein signaling protein 4 (RGS4). Furthermore, we characterized the effects of WIN55,212-2 on CB1-GIRK1/2-RGS4, CB2-GIRK1/2-RGS4, and GIRK1/2 systems, and compared the effects of several typical cannabinoids. In consideration of the high similarity of inward rectifier K^+^ (IRK) channels to GIRK channels and wide expression as well as vital functions of IRK1 (Kir2.1) in the human body, the effects of WIN55,212-2 were also tested on an IRK1-expression *Xenopus* oocyte system. This assay provides a straightforward and efficient methodology for examining cannabinoid-stimulated effects. Moreover, the results may partly explain the contradictory effects produced by WIN55,212-2 *in vivo*.

## 2. Materials and Methods

### 2.1. Isolation of Xenopus Oocytes

All procedures for the use and handling of adult female *Xenopus laevis* frogs (CRB Xénopes, Rennes, France) were approved by the Animal Ethics Committee of the KU Leuven (Project No. P186/2019) following regulations of the European Union (EU) concerning the welfare of laboratory animals as declared in Directive 2010/63/EU. Oocytes were isolated from ovarian tissue surgically removed during hypothermia and 0.1% tricaine (Sigma-Aldrich Chemical, St. Louis, MO, USA) induced anesthesia. After recovery from anesthesia, frogs were returned to their tanks in the Aquatic Facility of KU Leuven and monitored daily.

The oocytes were enzymatically defolliculated by collagenase (3 mg/mL) (Sigma-Aldrich Chemical, St. Louis, MO, USA) digestion at 16 ℃ on a rocker platform in a Ca^2+^-free ND96 solution. Isolated stage V-VI oocytes were then maintained in ND96 solution containing Theofylline and Gentamicin at 16 °C. The ND96 solution was composed of 96 mM NaCl, 2 mM MgCl_2_, 2 mM KCl, 5 mM HEPES, and 1.8 mM CaCl_2_, with a final pH of 7.5.

### 2.2. Heterologous Expression in Xenopus Oocytes

Linearized cDNAs of human CB1, CB2, GIRK1, GIRK2, IRK1, and RGS4 were used to transcribe cRNAs *in vitro* using the appropriate RNA polymerase (T3, SP6, or T7). Among these cDNAs, CB1, CB2, IRK1, and RGS4 cDNAs contain pGEMHE vectors which were constructed as described before [28], containing 5′ and 3′ untranslated regions of the *Xenopus* β-globin gene that flanks the cloned cDNA and enhances RNA stability in *Xenopus* oocytes [28]. The concentrations and quality of the cRNAs were determined by spectrophotometric absorbance at 260 nm and 280 nm on a spectrophotometer (NanoDrop ND-1000 UV/Vis, Delaware, USA). On the first day after enzymatic isolation (day 1), oocytes were injected with a mixture of cRNAs dissolved in nuclease-free water at a final injection volume of 50 nL (Nanoliter Injector A203XVZ, World Precision Instruments, Sarasota, FL, USA). Oocytes were injected with cRNAs of CB1 or CB2 (50–75 ng), GIRK1 (50–75 ng), GIRK2 (50–75 ng), and RGS4 (50 ng) mixtures, GIRK1 (50–75 ng) and GIRK2 (50–75 ng) mixtures, or IRK1 (~50 ng) alone, and then recorded K^+^ currents on day 2 or 3 for GIRK1/2 and IRK1, and day 4 or 5 for CB-GIRK1/2-RGS4 coupling.

### 2.3. Electrophysiological Recordings

Macroscopic K^+^ currents through GIRK1/2 were measured using a two-electrode voltage-clamp amplifier (GeneClamp 500B, Axon Instruments, San Jose, CA, USA). Electrodes were fabricated from borosilicate glass tubes (1.14 mm outside diameter, 0.7 mm inside diameter) by a programmable microelectrode puller (PUL-1, World Precision Instruments, Sarasota, FL, USA). Electrodes were filled with 3 M KCl and had tip resistances of 0.8 to 1.5 MΩ. Membrane currents from voltage-clamped oocytes were digitized using a Digidata 1550 low-noise data acquisition system (Axon Instruments, San Jose, CA, USA) and a Dell PC running pCLAMP 10.1 software (Axon Instruments, San Jose, CA, USA).

Oocytes were placed in a 0.2 mL recording chamber continuously perfused with ND96 solution. After electrode impalement and clamping the membrane potential to −90 mV, the perfusion solution was changed to a high K^+^ (HK) solution composed of 96 mM KCl, 2 mM NaCl, 1 mM MgCl_2_, 1.8 mM CaCl_2_, 5 mM HEPES with a final pH of 7.5. The resulting increase in inward K^+^ current represents a ‘basal’ K^+^ current (I_K,basal_) that is following primarily receptor-independent GIRK channel activity. Rapid application and washout of CB agonist WIN 55,212-2 (Sigma-Aldrich Chemical, St. Louis, MO, USA) in HK produced the reversible receptor-dependent K^+^ current (I_K,WIN_) and was performed with a controlled perfusion system that rapidly switched the perfusion of two syringe reservoirs (syringe A and syringe B) connected to syringe needles, two-way stopcocks, and tubing. The syringe reservoirs contained HK (syringe A) and HK plus WIN55,212-2 (syringe B). The perfusion system was located next to the oocyte. In addition to the abovementioned −90 mV voltage protocol, a 1-s voltage ramp protocol was also applied from −150 to +60 mV from a holding potential of −20 mV during perfusion.

For experiments to determine the concentration-current response relationship, a range of WIN 55,212-2 concentrations (in HK) were tested for each oocyte via a manifold that connected multiple syringe reservoirs containing different concentrations of WIN55,212-2. DMSO in each WIN55,212-2 solution was ≤1% (*v*/*v*), which did not cause changes of currents in non-injected oocytes, or oocytes expressing IRK1, GIRK1/2, and CB-GIRK1/2-RGS4. The 50 μM and 100 μM concentrations of WIN55,212-2 almost reached saturation of WIN55,212-2 in HK, indicated as ≈50 μM and ≈100 μM. Flow, through the perfusion system, was gravity-driven. All recordings were performed at room temperature (21 to 23 °C).

### 2.4. Electrophysiological Data Analysis

The EC_50_ values with corresponding 95% confidence intervals (CIs) were calculated using the Prism 8.0 program (GraphPad, San Diego, CA, USA). Statistical comparisons between the various experimental groups were performed by one-way ANOVA where *p* < 0.05 was considered significant. Experiments were replicated for three to six oocytes (*n* = 3–6).

## 3. Results

### 3.1. Activation of CB-GIRK1/2-RGS4 Coupling in Xenopus Laevis Oocytes

First of all, we aim to assess the properties of receptor-dependent GIRK1/2 activation elicited by cannabinoid receptors CB1 and CB2. In addition to cannabinoid receptors and GIRK1/2 in this system, RGS4 was also transfected to the oocytes. RGS proteins have been previously reported to maintain the robustness of GPCR-GIRK coupling [29]. Moreover, co-expression with RGS4 was beneficial to the constructs of CB-G(α)-G(βγ) in heterogeneous cells, stimulating the basal activity of cannabinoid receptors [30]. In this study, oocytes co-injected with CB1 or CB2, GIRK1/2, and RGS4 cRNAs were voltage-clamped at −90 mV. GIRK1/2 was initially inactivated by ND96, as shown in Figure 1A,B. After exchanging ND96 to HK, basal K^+^ currents, which are known to depend on free G protein βγ-subunits present in the oocytes because of the inherent activity of G proteins [31], were observed and were independent on the level of GIRK1/2 (I_K,basal_). In the presence of HK, K^+^ current enhancement was immediately evoked on the application of 1 μM WIN55,212-2 (I_K,WIN_) and was reversible via washing out WIN55,212-2 by HK. The receptor-dependent K^+^ currents represent that GIRK1/2 were activated by CB1 or CB2 coupled specifically to GIRK1/2. As a control, K^+^ currents carried by GIRK1/2 were not significantly enhanced upon the application of 1 μM WIN55,212-2 in oocytes injected with only GIRK1/2 (I_K,WIN_ = 0.00997 ± 0.0157 μA, *n* = 3).

To determine the intrinsic current (I_K,intrinsic_) that was responsible for I_K,basal_, non-injected oocytes were tested for sensitivity to HK. As a result, I_K,intrinsic_, which was 0.158±0.0595 μA (*n* = 5), constituted a negligible proportion of the total I_K,basal_. Exchanging HK to HK plus 1 μM WIN55,212-2 did not change I_K,intrinsic_ (*n* = 5), suggesting no effect of WIN55,212-2 on intrinsic oocyte receptors or other endogenous ion channels.

Furthermore, to study whether WIN55,212-2 can activate inward rectification of GIRK1/2 through activating CB1 and CB2, a voltage ramp protocol was applied. Oocytes co-injected with CB1 or CB2, GIRK1/2, and RGS4 cRNAs were subjected to a 1-s voltage ramp protocol from −150 to +60 mV from a holding potential of −20 mV during perfusion. Figure 2 is a representative example of these experiments. Inward rectifying currents, activated under the conditions of HK or HK plus WIN55,212-2, had a reversal potential near 0 mV, as I_K,in_ and I_K,out_ were approximately equal. The 1 μM WIN55,212-2 did not shift the K^+^ equilibrium potential of GIRK1/2, but simply increased the amplitude of inward K^+^ currents induced by HK. Thus, the current enhancement was presumably a result of increased GIRK1/2 channel conductance.

### 3.2. Enhancement and Reduction of Inward K^+^ Currents by WIN55,212-2 in a CB-GIRK1/2-RGS4 Coupling Oocyte Expression System

We investigated the concentration-response relations of the effects of WIN55,212-2 on the CB-GIRK1/2-RGS4 coupling expressed in *Xenopus* oocytes. Oocytes co-injected with CB1 or CB2, GIRK1/2, and RGS4 cRNAs were subjected to the voltage ramp protocol abovementioned in “3.1”. I_K,basal_ was first allowed to reach a steady-state when perfusing HK. Afterward, HK plus increasing concentrations of WIN55,212-2 (0.01–≈100 μM) was applied to independent oocytes (three oocytes per concentration). Note the progressively higher response of I_K,WIN_ with higher WIN55,212-2 concentrations (Figure 3). However, when the concentration reached ≈50 μM for the CB1-GIRK1/2-RGS4 coupling, the response of I_K,WIN_ was 50.9% less than that produced by 10 μM WIN55,212-2 (Figure 3). For the CB2-GIRK1/2-RGS4 coupling, 5 μM WIN55,212-2 already slightly reduced inward K^+^ currents (Figure 3). Subsequently, the extent of reduction of inward K^+^ currents increased with much higher concentrations of WIN55,212-2 in a concentration-dependent manner (≈100 μM for CB1-GIRK1/2-RGS4, and 5–≈100 μM for CB2-GIRK1/2-RGS4). Figure 4A,B show examples of the reduction in inward K^+^ currents produced by ≈100 μM WIN55,212-2 in the CB-GIRK1/2-RGS4 coupling expression system.

Plotting the percentage of current enhancement and reduction (I_K,WIN_/I_K,basal_ × 100) as a function of the WIN55,212-2 concentration reveals a concentration-dependent increase and subsequent decrease in the response of I_K,WIN_ (Figure 3).

For the increase section of the concentration–response curve, the best-fit indicates an apparent EC_50_ value of 0.374 μM (95% CI: 0.276 to 0.504) for activating CB1 (*n* = 3; Figure 5A), while the best-fit indicates an apparent EC_50_ value of 0.260 μM (95% CI: 0.163 to 0.414) for activating CB2 (*n* = 3; Figure 5B). The EC_50_ value for activating CB1 is consistent with previous studies where EC_50_ values were shown to range from 0.03 μM to 3 μM depending on the CB1 expression systems [1].

### 3.3. Blockage of GIRK1/2 via High Concentrations of WIN55,212-2

WIN55,212-2 is a well-characterized agonist of CB1 and CB2; thus, it can exert activity by activating cannabinoid receptors. Moreover, low concentrations of WIN 55,212-2 (≤1 μM) can activate receptor-dependent inward K^+^ current in CB1 or CB2 -GIRK1/2-RGS4 coupling oocyte expression system as described above, which is consistent with its agonistic activity. To investigate whether high concentrations of WIN55,212-2 interact with brain-type GIRK1/2, concentrations ranging from 1 to ≈100 μM WIN55,212-2 in HK were applied to oocytes injected with only GIRK1/2 cRNAs. The 1-s ramp protocol was also applied in this experiment. After reaching a steady-state of I_K,basal_ when perfusing HK, the application of 1 μM WIN55,212-2 plus HK still did not cause a significant effect on the inward K^+^ currents (*n* = 5, data not shown). Nonetheless, as shown in Figure 3 and Figure 4C, the application of higher concentrations (10–≈100 μM) of WIN55,212-2 plus HK to GIRK1/2-expressed oocytes caused a reduction of inward K^+^ currents in a concentration-dependent manner, which indicates high concentrations of WIN55,212-2 blocked GIRK1/2. Therefore, the reduction of inward K^+^ currents produced by WIN55,212-2 observed in the results of “3.2” was caused by the blockage of GIRK1/2 in the CB-GIRK1/2-RGS4 coupling expression system. In other words, these results suggest that high concentrations of WIN55,212-2 can block the effect of GIRK1/2 activated by CB1 and CB2, presumably following direct interaction with the GIRK channels.

GIRK2 can also form homomultimers in the human body [32,33] in addition to heterotetramer GIRK1/2. To further address the relation between WIN55,212-2 and GIRK1/2, we investigated the effects of 10 μM on GIRK2. In oocytes expressing the homotetramer GIRK2, 10 μM WIN55,212-2 blocked the homotetramer GIRK2 to half extent (8.91 ± 1.68% reduction in inward K^+^ currents, *n* = 3) when compared with heterotetramer GIRK1/2 (18.2 ± 1.40% reduction in inward K^+^ currents, *n* = 3).

### 3.4. Comparison of Effects of Other Typical Cannabinoid Members on GIRK1/2

We studied whether Δ9-THC (a partial agonist of CB1 and CB2), cannabidiol (CBD, an antagonist/inverse agonist or a negative allosteric modulator of CB1 and a partial agonist of CB2), CP55,940 (a non-selective agonist of CB1 and CB2), and rimonabant (a selective antagonist of CB1) also interact with GIRK1/2. Among these cannabinoids, Δ9-THC and CBD are derived from the plant *Cannabis sativa*, commonly known as marijuana. They are phytocannabinoids that have been extensively investigated and are classified as classical cannabinoids at the structural level [34] (Figure 6(B(a),C(a))). The classical group consists of dibenzopyran derivatives that are either cannabis-derived compounds (phytocannabinoids) or their synthetic analogs [34]. However, WIN55,212-2, CP55,940, and rimonabant are chemically synthesized, representing synthetic cannabinoids with three different chemical structures. The structure of WIN55,212-2 is the amino-alkylindole type and CP55,940 belongs to the non-classical type group which consists of bicyclic and tricyclic analogs of ∆9-THC that lack a pyran ring [34] (Figure 6(A(a),D(a))). Rimonabant belongs to the group of other types that have structures completely different from those of the abovementioned cannabinoids (Figure 6(E(a))). In this experiment, large inward K^+^ currents were also observed when the oocytes expressing GIRK1/2 were perfused with HK. Interestingly, the addition of 100 μM Δ9-THC to HK slightly enhanced the response of inward K^+^ currents (Figure 6(B(b))), whereas the addition of 100 μM CBD, CP55,940, or rimonabant to HK slightly reduced inward K^+^ currents (Figure 6(C(b),D(b),E(b))). Comparison of their effects with that of ≈100 μM WIN55,212-2 is shown in Figure 7. These results indicate that the ability of WIN55,212-2 to block GIRK1/2 is stronger than that of CBD, CP55,940, and rimonabant, while 100 μM Δ9-THC can modestly activate GIRK1/2.

The structure of all human Kir channels is very similar. They are all composed of four subunits that are each formed from two membrane-spanning helices connected by an extracellular domain and a pore helix, and cytosolic N- and C-termini [35]. Moreover, the homology among human Kir channels is relatively high with an identity that ranges between ∼28% (between Kir7.1 and Kir6.1 or Kir6.2) and ∼70% (between IRK1 and IRK2, and between GIRK2 and GIRK4) [35]. In particular, the identity between human IRK1 and GIRK1 is 42% [36], and the identity between human IRK1 and GIRK2 is ∼45% [35]. Additionally, IRK1 was found to be widely expressed in the human body [37] and is an important contributor to the resting membrane potential of the cells [38]. To explore the selectivity of the effect of WIN55,212-2, we tested whether the high concentration (≈100 μM) of WIN55, 212-2 interacts with IRK1, a constitutively active Kir channel. Intriguingly, in oocytes expressing IRK1, the application of ≈100 μM WIN55,212-2 in HK did not cause a significant effect on inward K^+^ currents through the GIRK1/2 (Figure 6(c)). The other typical cannabinoids Δ9-THC, CBD, CP55,940, and rimonabant in the presence of HK also did not produce obvious effects on inward K^+^ currents (Figure 6(c)).

## 4. Discussion

Cannabinoid receptors have been expectantly viewed as hopeful targets for the treatment of diverse diseases. Therefore, many expression systems of CB1 and CB2 have emerged in response to promptly discovering drug candidates by screening potential ligands *in vitro*, e.g., mammalian cells transfected with human cannabinoid receptors, certain cultured cell lines that express cannabinoid receptors naturally, and cannabinoid receptor-containing membrane preparations obtained from tissues (such as brain and spleen) [1,39]. The CB-GIRK1/2-RGS4 oocyte expression system built in this research is also a robust and reliable system for rapidly analyzing functions of cannabinoids or potential cannabinoids. Using this system, it is possible to investigate the effects of cannabinoids or potential cannabinoids on CB1 and/or CB2 in combination with their accessory functional proteins, e.g., GIRK channels. As mentioned in “1. Introduction”, GIRK channels are directly opened by Gi/o(βγ) subunits which are dissociated from Gi/o(α) subunits via activating cannabinoid receptors. Thus, the expression system of cannabinoid receptors used in this research nicely mimics the CB-GIRK signaling pathway. The other component, RGS4, in the expression system maintained the robustness of CB-GIRK1/2 coupling. RGS proteins are key components of GPCR complexes, interacting directly with G protein α-subunits to enhance their intrinsic GTPase activity [40]. Among RGS proteins composed of more than 30 genes, RGS4 expression has been found to markedly enhance Gi/o-coupled receptor-activated GIRK currents in *Xenopus* oocytes [29]. This study also reports that RGS4 augmented GIRK currents by increasing the availability of free Gi/o(βγ) rather than directly activating GIRK channels or potentiating Gi/o(βγ)-mediated gating of GIRK channels [29]. Furthermore, RGS4 dramatically improved the sensitivity of the CB-Gα-Gβγ-RGS4 construct system in Sf9 insect cells for screening potential ligands [30]. Thus, CB2-GIRK1/2-RGS4 was successfully constructed in this research, presumably because of the presence of RGS4 in a *Xenopus* oocyte expression system, compared with the previously built system [27]. Another advantage of using the heterologous *Xenopus* oocyte expression system is that different combinations of membrane-bound proteins can be freely expressed: for example, co-inject mRNA encoding CB1 and TRPV1 (the vanilloid receptor, also the target of some cannabinoid ligands), or CB2 and μ-opioid receptors, to examine the possible promiscuity of ligands, or CB1 with different types of voltage-gated ion channels (e.g., Na, K, Ca, Cl). Besides, structure-function research, based on the expression of mutant CB1/CB2, can also be conducted using this system. As such, this functional bioassay can unravel hitherto unknown coupling(s) of signal-transmission pathways, e.g., the cannabinoid and opioid pathways, allowing us to better understand the properties of some cannabinoids [1]. In addition to orthosteric ligands, discovery and characterization of allosteric modulators can also be achieved using this system, thus, functionally and structurally exploring allosteric modulation of CB receptors.

Interestingly, dual modulation of CB-GIRK1/2-RGS4 coupling was elicited by WIN55,212-2 in the oocyte expression system. As a non-selective agonist of CB1 and CB2, WIN55,212-2 in HK (≤1 μM) undoubtedly augmented inward K^+^ currents by activating CB1 or CB2. However, we further found that high concentrations of WIN55,212-2 (≈50 μM and ≈100 μM in CB1-GIRK1/2-RGS4 coupling, 5-≈100 μM in CB2-GIRK1/2-RGS4 coupling) in HK blocked GIRK1/2 activated by CB1 or CB2. Concentration-response curves of WIN55,212-2 are shown in Figure 3. The decline of the curve for CB1-GIRK1/2-RGS4 coupling is slower than that for CB2-GIRK1/2-RGS4 coupling and GIRK1/2. For instance, in the CB1-GIRK1/2-RGS4 coupling, 10 μM WIN55,212-2 in HK still dramatically enhanced inward K^+^ currents. In contrast, in the CB2-GIRK1/2-RGS4 coupling and GIRK1/2, 10 μM WIN55,212-2 in HK slightly reduced inward K^+^ currents. Thus, we consider that the affinity of 10 μM WIN55,212-2 for CB1 may be stronger than that for CB2 and GIRK1/2. Moreover, ≈50 μM WIN55,212-2 probably shared a similar affinity for both CB1 and GIRK1/2, while the affinity of ≈100 μM WIN55,212-2 for GIRK1/2 may become stronger than that for CB1 in the CB1-GIRK1/2-RGS4 coupling (Figure 3). Nevertheless, in the CB2-GIRK1/2-RGS4 coupling, 5–≈100 μM of WIN55,212-2 may primarily bind to GIRK1/2. Previous studies have reported that K_i_ (the binding constant) values for CB1 ranged from 1.89 to 123 nM, and the K_i_ values for CB2 varied from 0.280 to 16.2 nM, depending on various CB1 and CB2 expression systems [41]. Therefore, it is likely that the affinity of WIN55,212-2 under certain concentrations (5–≈50 μM) for CB1 is stronger than that for CB2. Additionally, a stronger affinity of WIN55,212-2 for GIRK1/2 is probably with higher WIN55,212-2 concentrations in CB-GIRK1/2-RGS4 coupling. These possibilities warrant, however, further systematic investigation. Exploring the relationship between GIRK1/2 and WIN55,212-2 further revealed that high concentration (10μM) of WIN55,212-2 also blocks homomultimer GIRK2. The extent of blockage is almost half of the extent blocking heterotetramer GIRK1/2. We speculate that high concentrations of WIN55,212-2 can bind to and block both GIRK1 and GIRK2 in heterotetramer GIRK1/2.

Functionally, WIN55,212-2 under high concentrations significantly blocked GIRK1/2 activated by CB1 or CB2, and thus inhibited the CB-GIRK1/2 signaling. Previous studies have shown that after chronic administration of WIN55,212-2 to adult male rats, CB1 binding and mRNA levels were reduced in the substantia nigra and the striatum [42]. Moreover, chronic treatment with WIN55,212-2 impaired the recognition memory and brain network functional connectivity of adult male mice [26]. Also, chronic exposure to WIN55,212-2 during adolescence induced persistent emotional and cognitive dysfunctions that reflect some facets of schizophrenia disease, including increased anxiety and alterations in spatial and episodic-like memory [43,44]. It is notable that WIN55,212-2 disrupted sensorimotor gating and attentional filtering in adolescent animals and that these disruptions were selectively reversed by antipsychotics such as haloperidol and risperidone [45,46,47]. These behavioral impairments associated with chronic exposure to cannabinoids during adolescence were mainly related to alterations in dopaminergic neurotransmission in the prefrontal cortex, which is a prominent component of cognitive dysfunctions in schizophrenia disease [48,49]. Furthermore, the frequency and amplitude of transient dopamine release events increased as WIN55,212-2 was administered in a cumulative, ascending manner for a chronic vehicle in rats [50]. As described in “1. Introduction”, the release of neurotransmitters including dopamine, i.e., neurotransmission, can be inhibited by activating GIRK channels. Additionally, mice lacking GIRK2 exhibited dopamine-dependent hyperactivity and elevated responses to drugs that stimulate dopamine neurotransmission [51]. Thus, the increased dopamine-releasing and altered neurotransmission in brain regions under chronic administration of WIN55,212-2 observed in the above studies might be because of the blockage of GIRK1/2 produced by accumulated high concentrations of WIN55,212-2, as shown in the present study.

Heavy or regular cannabis/cannabinoid abuse, generally defined as daily or almost daily use over a prolonged period, has been linked to cognitive dysfunction and increased risk of developing psychiatric symptoms, including schizophrenia spectrum disorders, acute psychosis, mania, and a motivational syndrome [26]. Nevertheless, no available data shows the concentrations of WIN55,212-2 after prolonged/chronic treatment *in vivo*. Similarly, the concentrations of Δ9-THC *in vivo* are, as far as we know, also unknown after chronic exposure of single Δ9-THC, though many studies have revealed the plasma concentration of Δ9-THC in humans after chronic treatment of cannabis. Meanwhile, cannabis/cannabinoid-based medicines are increasingly being used to treat several diseases such as epilepsy, chronic pain, multiple sclerosis, and neurodegenerative diseases [52,53,54,55], but the potential for negative side effects has not been well characterized. Understanding the effects of chronic cannabinoid exposure on brain and synaptic function will open a window into the development of therapeutic tools that could counteract the ‘on target’ side effects associated with chronic use of cannabis and cannabinoid-based medicines [56,57].

Finally, we compared the effects of ≈100 μM WIN55,212-2 with the effects of 100 μM of other typical cannabinoids Δ9-THC, CBD, CP55,940, and rimonabant on GIRK1/2. As shown in Figure 6((b),(c)) and Figure 7, only WIN55,212-2 significantly reduced inward K^+^ currents under the high concentration. This result may be attributed to the amino-alkylindole-type structure of WIN55,212-2 which is distinct from the structures of other tested cannabinoids. On the other hand, human IRK1 was not sensitive to high concentrations of WIN55,212-2, Δ9-THC, CBD, CP55,940, and rimonabant. The core cytoplasmic domain elements of GIRK1 or GIRK2 channels are very similar to that of IRK1 [58]. The binding of G proteins triggers a conformational change of the cytoplasmic domain of GIRK channels [59]. The strong rectification of IRK1 has been attributed to three principal electronegative regions: Asp172 in the M2 domain, Glu224/Glu299 in the cytoplasmic domains, and di-aspartate cluster (Arg255/Arg259) in the cytoplasmic pore [60]. We hypothesize that WIN55,212-2 does not bind to these regions in IRK1, and thus cannot trigger inward rectification.

## 5. Conclusions

The present research built and validated a reliable and efficient CB1 and CB2 expression system coupled to GIRK1/2 and RGS4 in *Xenopus* oocytes. The CB2-GIRK1/2-RGS4 coupling system is the first successful heterologous expression system of CB2. This research firstly reveals the dual modulation of cannabinoid receptors and brain GIRK1/2 coupling by WIN55,212-2 in the oocyte expression system. Although WIN55,212-2 is an agonist of CB1 and CB2 and can activate CB-GIRK signaling in the CB-GIRK1/2-RGS4 coupling, it can also block GIRK1/2 activated by CB1 and CB2 under high concentrations. Moreover, WIN55,212-2 also can directly block GIRK1/2 in a concentration-dependent manner. Comparing the effects of WIN55,212-2 with the effects of Δ9-THC, CBD, CP55,940, and rimonabant which are also typical cannabinoids on GIRK1/2 shows that only WIN55,212-2 can significantly block GIRK1/2 at the high concentration. However, the high concentration of WIN55,212-2, Δ9-THC, CBD, CP55,940, and rimonabant did not affect IRK1, which is highly similar to GIRK channels, representing the selective blockage of WIN55,212-2 on GIRK1/2. The blockage of GIRK1/2 in CB-GIRK signaling provides new insights on the cannabinoid receptors-mediated side effects induced by the chronic accumulation of WIN55,212-2 *in vivo*. This research also provides a straightforward methodology to rapidly test the effects of cannabinoids/potential cannabinoids on CB1 and CB2 *ex vivo*.

## Figures and Tables

**Figure 1 biomedicines-09-00484-f001:**
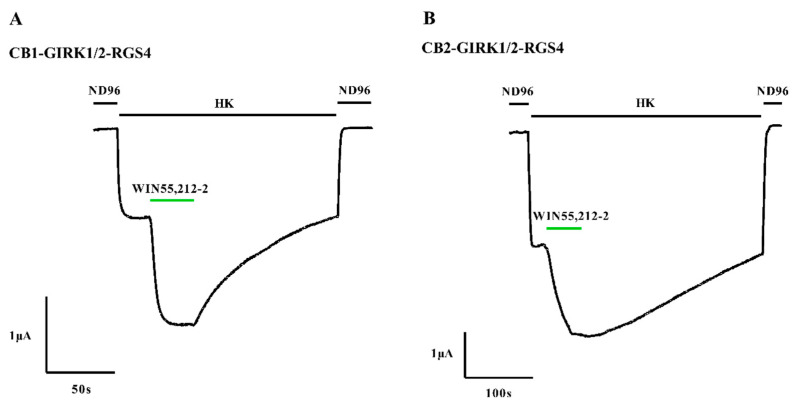
Activation of the CB-GIRK1/2-RGS4 coupling in *Xenopus* oocytes. (**A**) Oocytes co-injected with CB1, GIRK1/2, and RGS4, and (**B**) oocytes co-injected with CB2, GIRK1/2, and RGS4 were voltage-clamped at −90 mV. I_K,basal_ was enhanced by exchanging HK to HK plus 1 μM WIN55,212-2. Typical I_K,basal_ and I_K,WIN_ were recorded from three to six different oocytes (*n* = 3–6).

**Figure 2 biomedicines-09-00484-f002:**
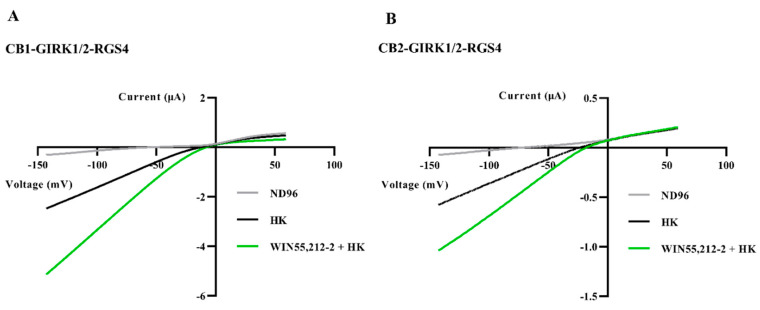
Activation of inward rectification of GIRK1/2 in the CB-GIRK1/2-RGS4 coupling in *Xenopus* oocytes. (**A**) Oocytes co-injected with CB1, GIRK1/2, and RGS4 cRNAs, and (**B**) oocytes co-injected with CB2, GIRK1/2, and RGS4 cRNAs were subjected to a 1-s voltage ramp protocol from −150 to +60 mV from a holding potential of −20 mV. Inward I_K,basal_ was enhanced by exchanging HK to HK plus 1 μM WIN55,212-2. Typical I_K,basal_ and I_K,WIN_ were recorded from three to six different oocytes.

**Figure 3 biomedicines-09-00484-f003:**
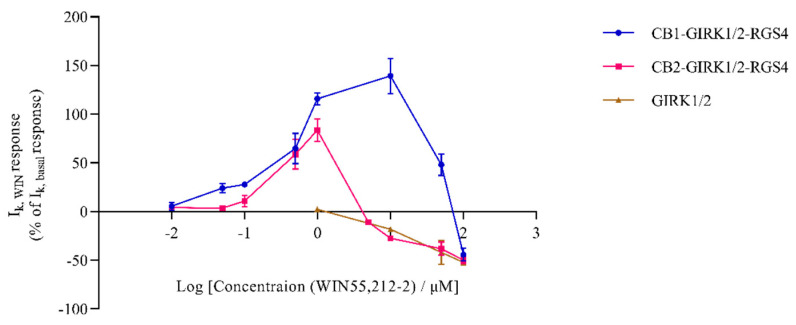
Concentration-response curve of WIN55,212-2 for CB-GIRK1/2-RGS4 coupling and GIRK1/2. Oocytes were co-injected with CB, GIRK1/2, and RGS4, or GIRK1/2 cRNAs. Inward K^+^ current enhancement and reduction were produced on the application of a range of different concentrations of WIN55,212-2 in the presence of HK. The positive percentage represents enhancement and the negative percentage represents reduction. Each data point is the mean ± standard deviation (SD) of three determinations from two or three batches of oocytes.

**Figure 4 biomedicines-09-00484-f004:**
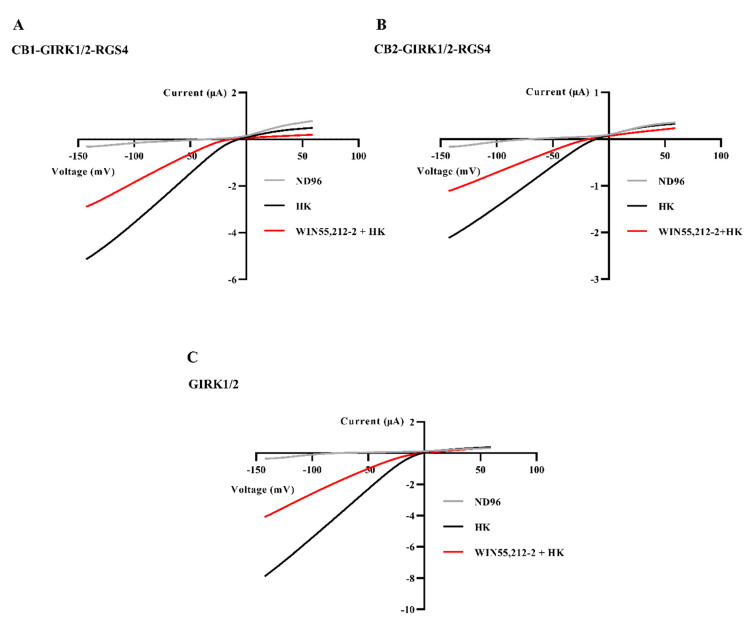
Reduction of inward K^+^ currents in the CB-GIRK1/2-RGS4 coupling and GIRK1/2 in *Xenopus* oocytes. (**A**) Oocytes co-injected with CB1, GIRK1/2, and RGS4 cRNAs, (**B**) oocytes co-injected with CB2, GIRK1/2, and RGS4 cRNAs, and (**C**) oocytes injected with GIRK1/2 cRNAs were subjected to a 1-s voltage ramp protocol from −150 to +60 mV from a holding potential of −20 mV. Inward I_K,basal_ was strongly reduced by exchanging HK to HK plus ≈ 100 μM WIN55,212-2. Typical I_K,basal_ and I_K,WIN_ were recorded from three to six different oocytes.

**Figure 5 biomedicines-09-00484-f005:**
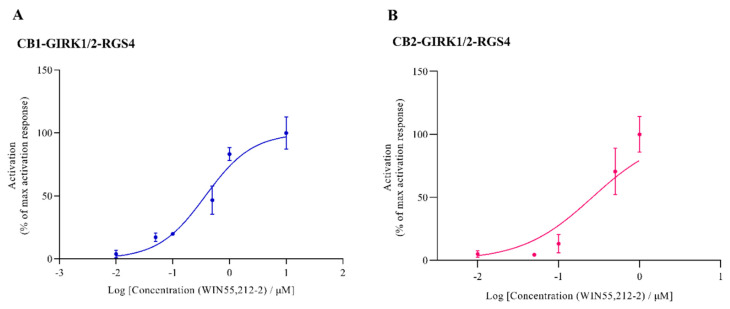
The best fit for concentration–current activation curve of WIN55,212-2 for CB-GIRK-RGS4 coupling. Oocytes were co-injected with CB, GIRK1/2, and RGS4 cRNAs. (**A**) CB1 activation and (**B**) CB2 activation were produced on the application of a range of different concentrations of WIN55,212-2 in the presence of HK. Each data point is the mean ± SD of three determinations from two or three batches of oocytes.

**Figure 6 biomedicines-09-00484-f006:**
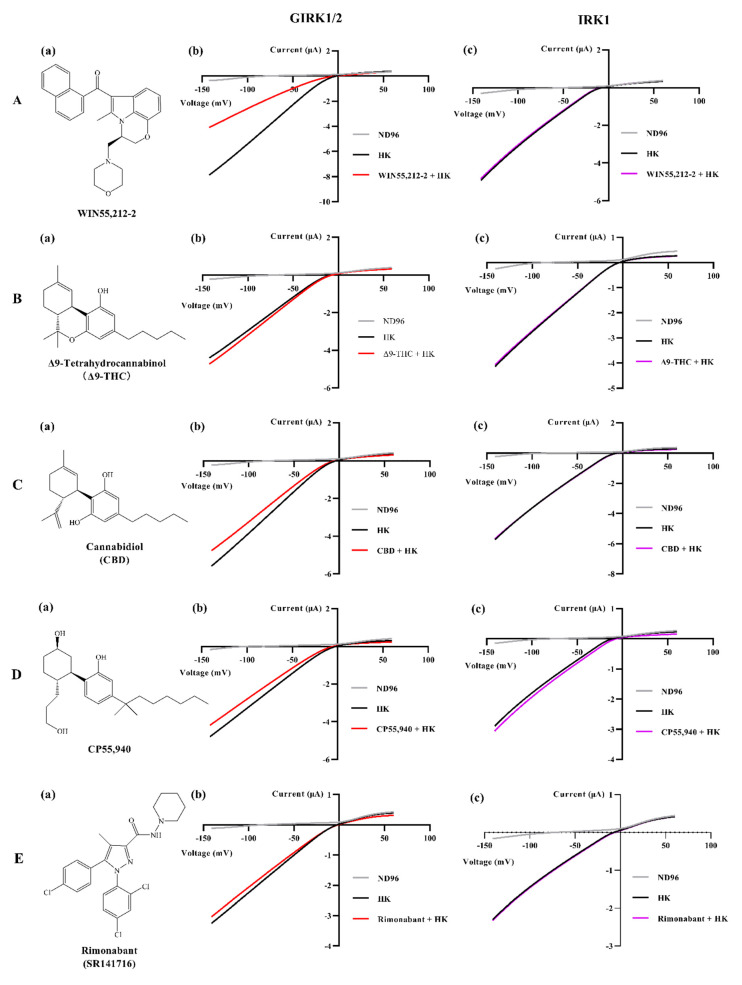
Structures of WIN55,212-2, ∆9-THC, CBD, CP55,940, and rimonabant, and changes of inward K^+^ currents through GIRK1/2 and IRK1 produced by these cannabinoids. Structures of WIN55,212-2, ∆9-THC, CBD, CP55,940, and rimonabant are shown in (**a**) column. (**b**) Oocytes injected with GIRK1/2 and (**c**) oocytes injected with IRK1 were subjected to a 1-s voltage ramp protocol from −150 to +60 mV from a holding potential of −20 mV. Current enhancement or reduction was produced on the application of ≈100 μM of (**A**) WIN55,212-2, or 100 μM of (**B**) ∆9-THC, (**C**) CBD, (**D**) CP55,940, and (**E**) rimonabant. Typical I_K,basal_ and I_K,compound_ were recorded from three to five different oocytes.

**Figure 7 biomedicines-09-00484-f007:**
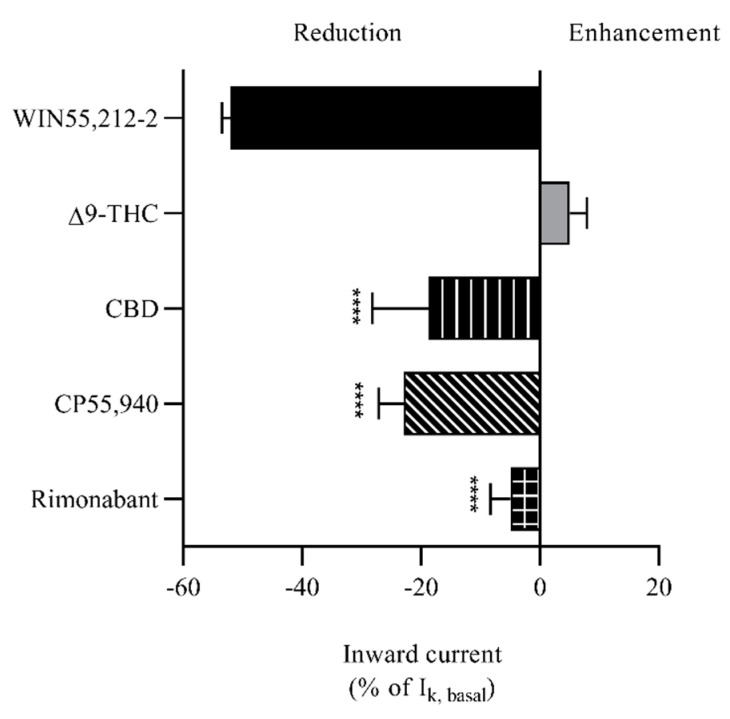
Comparison of effects of WIN55,212-2, ∆9-THC, CBD, CP55,940, and rimonabant on GIRK1/2. ≈100 μM of WIN55,212-2 and 100 μM of CBD, CP55,940, and rimonabant reduced inward K^+^ currents, while 100 μM ∆9-THC slightly enhanced inward K^+^ currents. The positive percentage represents enhancement and the negative percentage represents reduction. Statistical comparisons between the various experimental groups were performed by Dunnett’s multiple comparisons test of one-way ANOVA where *p* < 0.05 was considered significant. ****, *p* < 0.0001 vs. percentage for WIN55,212-2. Each data point is the mean ± SD of three determinations from two or three batches of oocytes.

## Data Availability

The data presented in this study are available on request from the corresponding author. The data are not publicly available due to privacy.

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
