# Peer review of "WIN55,212-2, a Dual Modulator of Cannabinoid Receptors and G Protein-Coupled Inward Rectifier Potassium Channels"

_biomedicines, 2021, doi:10.3390/biomedicines9050484_

Round 1
Reviewer 1 Report
The present study attests the functional expression in a Xenopus laevis oocyte expression system of CB1 or CB2 receptors, through interaction with heteromeric GIRK1 / 2 channels and a regulator of G protein signaling, RGS4. This ex vivo system made it possible to discover a large number of ligands interacting orthosterically or allosterically with CB1 and / or CB2 receptors. WIN55,212-2, it has been employed to investigate the involvement in the CB1 or CB2 signal cascade GIRK1/2-RGS4 and it has emerged that at low concentrations it activates CB1 and CB2 while at higher concentrations it shows a direct block of GIRK1/2. This double modulating function is important, it is a characteristic never shown that could support the adverse effects induced by WIN55,212-2 17 in vivo. Furthermore, after a comparison with other cannabinoids it was found that only WIN55,212-2 can significantly block GIRK1/2.
The paper is well written and text is clear to read.
The authors explored the topic and they obtained the purpose of the study.
The conclusions are consistent with the evidence and arguments presented.
Specific indications:
Regarding bibliographic references in the manuscript, in many works the publication date is not in bold as in the references 12, 13, 14, 16, 17, 22, 34, 35, 39, 40, 49 and 54.
Author Response
We thank the reviewer for the comments on our manuscript.
Following the reviewer’s remark, we have now corrected the references.
Reviewer 2 Report
The CB-GIRK1/2-RGS4 oocyte expression system appears to be generalizable in its results. The pharmacology, electrophysiology and statistics appear to be sound. While technical details appear solid, the authors could optionally expand the potential readers of this work with a paragraph in the discussion outlining future steps to generalize this system as it relates to in vivo or human cannabinoid research.
Author Response
We thank the review for the review of our manuscript and the useful suggestion to improve the text.
Following the reviewer’s remark, we have now added the following text to expand the potential readers of this work with a paragraph in the discussion outlining future steps to generalize this system for human cannabinoid research:
Line 365: Another advantage of using the heterologous Xenopus oocyte expression system is that different combinations of membrane-bound proteins can be freely expressed: for example, co-inject mRNA encoding CB1 and TRPV1 (the vanilloid receptor, also the target of some cannabinoid ligands), or CB2 and μ-opioid receptors, to examine the possible promiscuity of ligands, or CB1 with different types of voltage-gated ion channels (e.g. Na, K, Ca, Cl). Besides, structure-function research, based on the expression of mutant CB1/CB2, also can be conducted by using this system. As such, this functional bioassay can unravel hitherto unknown coupling(s) of signal-transmission pathways, e.g., the cannabinoid and opioid pathways, allowing us to better understand the properties of some cannabinoids [1]. In addition to orthosteric ligands, discovery and characterization of allosteric modulators also can be achieved via using this system, thus, functionally and structurally explore allosteric modulation of CB receptors.